# Time Series Analysis of Temperature and Rainfall in the Savannah Region in Togo, West Africa

Komlagan Mawuli Apelete Yao [1,*], Edinam Kola [2], Wole Morenikeji [3] and Walter Leal Filho [4]

1 West African Science Service Centre on Climate Change and Adapted Land Use, Accra CT 504, Ghana
2 Department of Geography, Faculty of Humanities and Social Sciences, University of Lomé, Lomé 01BP1515, Togo
3 Urban and Regional Planning Department, Federal University of Technology, Minna 920101, Nigeria
4 European School of Sustainability Science and Research, Hamburg University of Applied Sciences, Ulmenliet 20, 21033 Hamburg, Germany
* Correspondence: yao.k@edu.wascal.org

**Abstract:** This study investigates the trend in monthly and annual rainfall, and minimum and maximum temperature (Tmin and Tmax) in the Savannah region of Togo. The historical data of Mango and Dapaong weather stations from 1981 to 2019 were used. A serial correlation test was applied to all time series to identify serially independent series. A Non-parametric Mann–Kendall (MK) test was applied to serially independent series. The magnitude of the trend was calculated using the Sen's slope (SS) method. For the data influenced by serial correlation, a modified version of the Mann–Kendall test was applied. An open-source library package was developed in the R language, namely, "mkmodified". For annual rainfall, results showed a significant increasing trend at Dapaong ($p < 0.05$) and a non-significant decreasing trend at Mango ($p > 0.05$) at 95%. There was an increasing trend in the Tmin both at Mango and Dapaong. No statistically significant trend was found at Mango ($p > 0.05$), while at Dapaong ($p < 0.05$), a significant trend was found at 95%. Simlarly, there was a statistically increasing trend in the Tmax both at Mango and Dapaong. Rainfall in Dapaong has increased (5.50 mm/year) whereas in Mango, it has decreased ($-0.93$ mm/year). Tmn increased by 0.04 and 0.008 °C per year in Mango and Dapaong, respectively. Tmax increased by 0.03 and 0.02 °C per year in Mango and Dapaong, respectively. A Rainfall Anomaly Index (RAI) was also used to present a temporal variation in rainfall; the historical series presented drier years. Many studies have analysed the trend of climate parameters in northern Togo, but none of them has specifically targeted the Savannah region that is considered the poorest region of the country.

**Keywords:** trend analysis; Mann–Kendall test; Sen's slope estimator; temperature; rainfall

## 1. Introduction

Strong evidence and understanding exist that climate change is occurring, and it is acknowledged as one of the biggest concerns of our century. According to the IPCC (2007) [1], the temperature in Africa is predicted to increase by 1.5 to 3 °C by 2050. Malhi and Wright (2004) [2] discovered that African tropical forests and South Africa will get warmer by 0.29 °C. Adeola et al. (2022) [3], using a modified Mann–Kendall test on historical observation data from the period 1976–2019, found that the Olifants River Catchment in South Africa experienced an increase in temperature and an overall decline in rainfall, although no significant changes were detected in the distribution of rainfall time. Hulme et al. (2005) [4] stated that rainfall in Africa has both spatial and temporal variability: in West Africa since the late 1960s, rainfall has been decreasing (Nicholson and Selato, 2000; Chappell and Agnew 2004; Dai et al., 2004) [5–7] while the Guinean coast recorded a 10% increase in annual rainfall between 1995 and 2005. According to the IPCC (2007) [8], extreme occurrences such as droughts and floods are more often due to climate change, which also alters rainfall patterns. This is a fact that has an impact on rural livelihoods in West Africa and

is presenting a serious challenge for future development in the region (Lebel & Ali (2009); Jalloh et al. (2013) [9,10]). Using the non-parametric modified Mann–Kendall test on rainfall time series from 1940 to 2015 in Benin, Ahokpossi (2018) [11] found that no significant trend or breakpoint and changes in the variance were observed for the spatial average rainfall time series. From 1961 to 2000, the average temperature increased by 0.5 to 0.8 °C from south to north in Togo, and the number of rainy days from 2.22 to 3.3 mm annually to 10.6 to 14.4 days annually depending on the environment (Adewi et al., 2010) [12]. While the ratio of evapotranspiration to rainfall is below 0.75 in several places, the rising temperature exhibits a clear trend towards an arid climate (Amegadje, 2007) [13]. Similar to this, McSweeney et al. (2009) [14] found that the average annual temperature in Togo has risen by 0.24 °C since 1960, with annual rainfall being incredibly variable on both an annual and interdecadal timescale. The findings of Rehmani et al. (2015) [15] supported these conclusions. Djaman et al. (2017) [16] noted decreasing patterns in annual rainfall. Using the results of climate models, Jalloh et al. (2013) [10] cited in Gadedjisso (2021) [17] showed that Togo will experience a decrease in rainfall and an increase in temperature of between 1 °C and 2.5 °C by 2050. According to Somiyabalo (2019) [18], the climate in Northern Togo is also changing and fluctuating. Climate change is also pushing rural-to-urban migration in Togo to the edge of the absorptive capacity of urban centres, limiting their ability to manage adequately a growing flow of people in search of a better life. Direct consequences of harsher climatic conditions include the observed abandonment of rural areas correlated with lost interest in agricultural production and rapid depletion of natural resources, whereas urban expanses evidence the acceleration of informal settlement formation in slums with significant impacts on health, poverty, and social instability, which are also aggravated by climate change (Togolese Directorate of Environment, 2021). Most studies have considered the whole northern part of Togo when talking of the spatial–temporal distribution of temperature and rainfall. This study targeted particularly the Savannah region of Togo, because the region straddles the Sahelian zone and the tropical zone. As a result, the region's rainfall and temperature patterns are influenced by both the desert climate and the climate of Equatorial Guinea. Additionally, the region is the poorest in the country, and is experiencing serious climate variability.

Analysing rainfall patterns and variations based on historical data aids in a better understanding of the issues related to drought and flooding in relation with migration phenomenon. Therefore, using the modified Mann–Kendall test and Sen's slope, the current study explores the trend in monthly and yearly rainfall as well as in monthly and annual minimum and maximum temperatures in the Savannah region of Togo.

## 2. Materials and Methods

### 2.1. Description of the Study Area

Togo is a West African French-speaking country. The country is bordered by Ghana to the west and Benin to the east. Togo is located between the latitudes 6 N and 11 N, and longitude 0 E and 2 E. Its surface area is 56,600 km$^2$, and Togo has a long narrow profile, elongating more than 550 km from north to south but not greater than 160 km in width (Figure 1). The study area is in northern Togo. Historical climate observations were obtained from Dapaong and Mango meteorological stations because they are the only available meteorological stations in the study area.

### 2.2. Data Sources

For data quality, a range of years was chosen (1981 to 2019). Before been used for any analysis, the climate data used in this study were rigorously processed for missing values and quality confirmed. For the study's stations, there were no missing values in rainfall or temperature data from 1981 to 2019, Table 1.

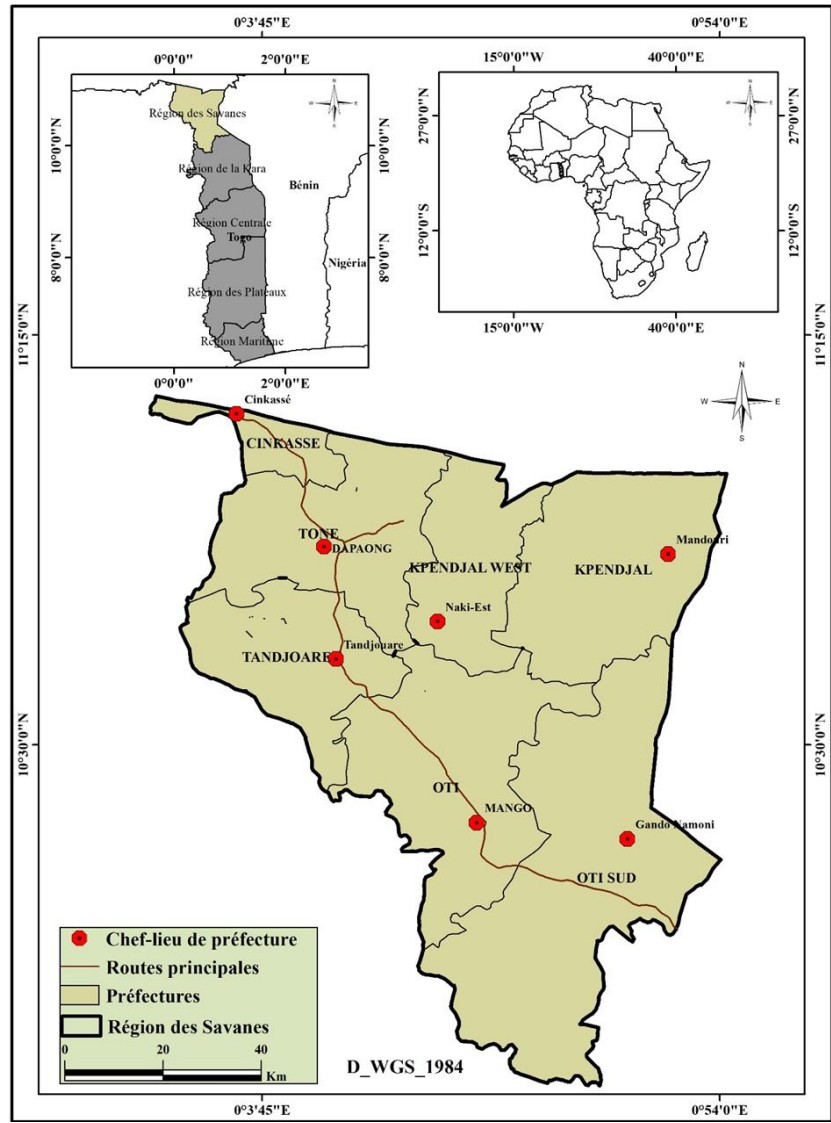

**Figure 1.** Map of the study site. Source: Hlovor, 2022.

**Table 1.** Characteristics of stations for recorded rainfall and temperature data in the Savannah region.

| Stations | Latitude | Longitude | Elevation AMSL * (m) | Study Period | Number of Years |
|---|---|---|---|---|---|
| Mango | 10.22 | 0.28 | 146 | 1981–2019 | 39 |
| Dapaong | 10.51 | 0.12 | 300 | 1981–2019 | 39 |

Notes: Source: General Directorate of National Meteorology, 2020. * Significant at 5%.

In this study, monthly and annual patterns of rainfall and temperature are examined for Mango and Dapaong weather stations. Daily rainfall and temperature records were obtained from the weather stations for the period of 1981 to 2019.

### 2.3. Testing for Serial Correlation

The statistical tests presume that the succeeding data in the series are independent in order to identify a trend in a time series. Serial correlation in the data has a significant impact on the power of trend tests (Yue et al. (2002); Onyutha (2016) [19,20]). When there is a trend, a positive serial correlation causes the null hypothesis of no trend to be incorrectly

rejected (Type I error). Similar to this, when there is a negative serial correlation, the null hypothesis of no trend is accepted even when it is wrong (Type II error). Lag$^{-1}$ serial correlation coefficients were generated (Kendall et al. (1968); Anderson (1942)) [21,22] to test for serial correlation in the data. The lag$^{-1}$ serial correlation coefficient (p1) is used in numerous trend studies to assess the time series for serial correlation (Chattopadhyay et al. (1942); Basistha et al. (2014)) [23,24].

For any time series $X_i = x_1, x_2, \ldots, x_n$, lag$^{-1}$ serial correlation coefficient ($P_1$) is calculated as

$$P_1 = \frac{\frac{1}{n-1}\sum_{i=1}^{n-1}(x_i - E(x_i))(x_{i+1} - E(x_i))}{\frac{1}{n}\sum_{i=1}^{n}(x_i - E(x_i))^2} \tag{1}$$

where $E(x_i)$ is the mean of the sample and $n$ is the sample size

$$E(x_i) = \frac{1}{n}\sum_{i=1}^{n}n \tag{2}$$

The probability limits for $P_1$ on the correlogram of an independent series is given by Anderson (1942) [5] as

$$P_1 = \begin{cases} \frac{-1+1.645\sqrt{n-2}}{n-1}, \text{ for the one}-\text{tailed test} \\ \frac{-1\pm1.96\sqrt{n-2}}{n-1}, \text{ for the two}-\text{tailed test} \end{cases} \tag{3}$$

Significance of serial correlation was evaluated by comparing the $P_1$ value with the critical values of Student's *t*-distribution.

*2.4. Trend Tests*

Time series data are subjected to a trend test to find significant positive or negative patterns. The Mann–Kendall test (Mann, 1945; Kendall, 1955) [25,26] was used to check for trends in the event that the time series are serially independent. This test is well known for spotting trends and is frequently used in conjunction with trends that have been predicted using Sen's slope approach (Sen, 1968) [27]. However, it has been highlighted in the literature that there is no way to account for the serial correlation that exists in time series, thus it is prudent to use some statistical tests to examine the current serially linked data. When a substantial serial correlation in the time series was found in this study, the Mann–Kendall test was applied: the Yue and Wang, (2004) [28] (MMKY) approach to variance correction.

2.4.1. Mann–Kendall Trend Test

In this particular study, the Mann–Kendall (MK) test was employed to assess the trends in rainfall and temperature. The non-parametric Mann–Kendall test does not require that the data be normally distributed and is less susceptible to outliers. The MK test is based on a null hypothesis (Ho), which states that there is no trend because the data are independent and randomly ordered. This null hypothesis is then checked against the alternative hypothesis (ha), which assumes that there is a trend. Sen's slope (SS) estimator was used to forecast the actual slope (change per unit time). The Mann–Kendall test was carried out using the Excel program.

$$S = \sum_{i-1}^{N-1}\sum_{j=i+1}^{n}sgn(xj - xi) \tag{4}$$

where $S$ is the Mann–Kendall test statistic; $xi$ and $xj$ are the sequential data values of the time series in the years $i$ and $j$ ($j > i$), and $N$ is the length of the time series.

$$sign(xj - xi) = \begin{cases} +1, (xj - xi) > 0 \\ 0, (xj - xi) = 0 \\ -1, (xj - xi) < 0 \end{cases} \tag{5}$$

In the data series, a positive $S$ value denotes an ascending trend while a negative value denotes a descending trend. For the case where there may be ties (i.e., equal values) in the $x$ values, the variance of $S$ is given by:

$$\text{Var}(S) = \frac{n(n-1)(2n+5) - \sum_{i=1}^{m} T_i i(i-1)(2i+5)}{18} \tag{6}$$

where $m$ is the number of tied groups in the data set and $T_i$ is the number of data points in the $i^{th}$ tied group. For $n$ larger than 10, $Zs$ approximates the standard normal distribution and computed as

$$Zs = \begin{cases} \frac{S-1}{\sqrt{var(S)}}, & if \ S > 0 \\ 0, & if \ S = 0 \\ \frac{S+1}{\sqrt{var(S)}}, & if \ S < 0 \end{cases} \tag{7}$$

The presence of a statistically significant trend is evaluated using the $Zs$ value. In a two-sided test for trend, the null hypothesis $H_0$ should be accepted if $|Zs| < z_{1-\frac{\alpha}{2}}$ at a given level of significance $\alpha = 0.05$. $|Z| < z_{1-\frac{\alpha}{2}}$ is the critical value of $Zs$ from the standard normal table; for the 5% significance level, the value of $z_{1-\frac{\alpha}{2}}$ is 1.96.

### 2.4.2. Sen's Slope Estimator Test

The magnitude of trends in the data time series was also estimated using the non-parametric technique given by Sen (1968). The following formula estimates the slope of "$n$" pairs of data:

$$\beta = \text{Median} \left( \frac{xj - xi}{j - i} \right) \ j > i \tag{8}$$

where $\beta$ is Sen's slope estimator, and $xj$ and $xi$ are data values at times $j$ and $i$, $(j > i)$, respectively. The "$n$" values of $\beta$ are ranked from the smallest to largest and the median of "$n$" values of $\beta$ is Sen's slope, which is given as

$$\begin{cases} \beta_{\left[\frac{(n+1)}{2}\right]} & If \ n \ is \ odd \\ \frac{1}{2}\left\{ \beta_{\left[\frac{n}{2}\right]}, + \beta_{\left[\frac{(n+2)}{2}\right]} \right\} & If \ n \ is \ even \end{cases} \tag{9}$$

A negative $\beta$ value indicates a declining tendency, while a positive $\beta$ value indicates an upward trend over time.

### 2.4.3. Variance Correction Approaches

Effective information is represented in $n*$ number of observations in a time series with $n$ observations where $X_t = x_1, x_2, \ldots, x_n$. The actual sample size n is never more than effective sample size $n*$. The variance of the Mann–Kendall test statistic $S$ changes depending on whether there is positive or negative serial correlation. The modified variance $V*$ can be calculated to lessen this effect ($S$).

$$V^*(S) = V(S) * CF \tag{10}$$

Correction factors ($CF$) proposed by Hamed and Rao (1998) [29] and Yue and Wang (2004) [30] known as $CF_1$ and $CF_2$, respectively, are as follows:

$$CF_1 = 1 + \frac{2}{n(n-1)(n-2)} \sum_{k=1}^{n-1} (n-k)(n-k-1)(n-k-2)r_k^R \tag{11}$$

$$CF_2 = 1 + 2\sum_{k=1}^{n-1} \left(1 - \frac{k}{n}\right) r_k \tag{12}$$

where $r_k$ and $r_k^R$ are the *lag—k* serial correlation coefficients of data and ranks of data, respectively, and *n* is the total length of the series. In the case of $CF_1$, only significant correlation coefficients were used. For calculating $CF_1$, the *lag—1* serial correlation coefficient was used. The Mann–Kendall trend test is calculated by using corrected variance $V^*(S)$.

*2.5. Variability Analysis*

2.5.1. Coefficient of Variation (CoV %)

Even when the means of two data series are very dissimilar from one another, the coefficient of variation, which measures the ratio of the standard deviation to the mean, can be used to compare the degree of variation between the two. The Coefficient of Variation, which is typically expressed as a percentage, is used to assess the variability of rainfall data in relation to its standard deviation and is normally presented as a percentage (Canchola et al., 2017) [31].

$$CoV = \frac{\sigma}{\mu} \tag{13}$$

where *CoV* is the Coefficient of Variation; $\sigma$ is the standard deviation, and $\mu$ is long term mean rainfall. According to Hare (2003), the values of *CoV* (<20) are considered less variable, (20–30) moderately variable, and (>30) highly variable.

Table 2 below is the classification of *CoV* values according to the American Society of Agriculture Engineere "ASAE" Standards.

**Table 2.** Interpretation of coefficient of variation.

| Emitter Type | Manufactures Coefficient | Interpretation |
|:---:|:---:|:---:|
| | <0.05 | Excellent |
| | 0.05–0.07 | Average |
| Point source | 0.07–0.011 | Marginal |
| | 0.11–0.15 | Poor |
| | >0.15 | Unacceptable |
| | <0.10 | Good |
| Line source | 0.10–0.20 | Average |
| | >0.20 | Marginal to |

Note: Source: ASAE/ASABE Standard 405.1, 2005.

2.5.2. Rainfall Anomaly Index (RAI)

The Annual Rainfall Anomaly Index (RAI) calculated from the precipitation data was used to determine the frequency and severity of the dry and rainy years in the research area. The difference between the yearly total of a given year and the long-term average rainfall records, divided by the standard deviation of the long-term data, is the rainfall anomaly index (Table 3), which is used to examine the frequency and intensity of the dry and rainy years in the past. Positive rainfall anomalies signal a year with more rain than the long-term average, whereas negative anomalies signal a year with less rain than the long-term average (a dry year). RAI is one of the major drought indices and is widely used to analyse the drought severity, intensity, and frequency. RAI has two anomalies: positive anomalies and negative anomalies. RAI, developed and first used by Rooy (1965) [31] and adapted by Freitas (2005) [32], constitutes the following:

$$RAI = 3\left[\frac{\bar{N} - N}{\bar{M} - \bar{N}}\right] \quad \textit{For positive anomalies} \tag{14}$$

$$RAI = 3\left[\frac{N - \bar{N}}{\bar{X} - \bar{N}}\right] \quad \textit{For negative anomalies} \quad (15)$$

where: $N$ = current monthly/yearly rainfall, in order words, of the month/year when *RAI* will be generated; (mm); $N$ = monthly/yearly average rainfall of the historical series (mm); $M$ = average of the ten highest monthly/yearly precipitations of the historical series (mm); $X$ = average of the ten lowest monthly/yearly precipitations of the historical series (mm); and positive anomalies have their values above average and negative anomalies have their values below average.

**Table 3.** Classification of Rainfall Anomaly Index Intensity.

|  | RAI Range | Classification |
|---|---|---|
| | Above 4 | Extremely humid |
| | 2 to 4 | Very humid |
| | 0 to 2 | Humid |
| Rainfall Anomaly Index (RAI) | −2 to 0 | Dry |
| | −4 to −2 | Very dry |
| | Below −4 | Extremely dry |

Note: Source: Freitas (2005) adapted by Araújo et al. (2009) [32,33].

## 3. Results and Discussion

### 3.1. Annual and Seasonal Rainfall and Temperature Trend Analysis

#### 3.1.1. Serial Correlation Analysis

The results of this study indicated that at Mango, only 1 series was found to be serially correlated. The critical value is ±0.347 for any series with 39 observations (Figure 2).

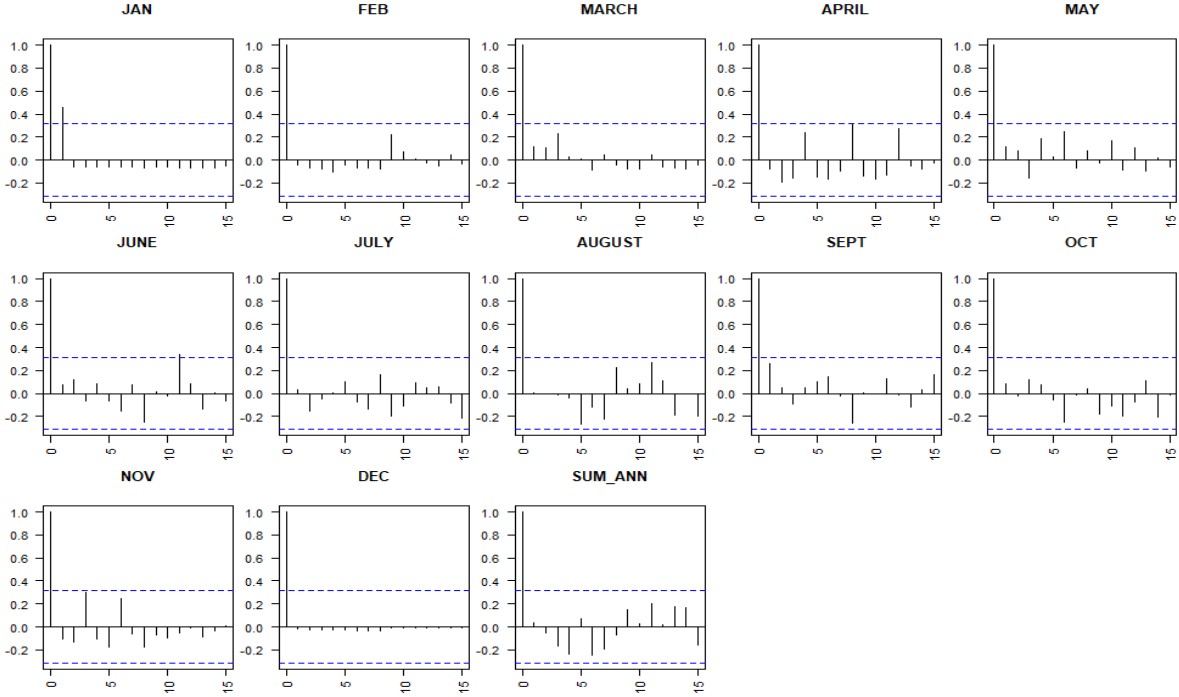

**Figure 2.** Autocorrelation test on monthly and annual rainfall (Mango station). Source: Mango meteorological service, 2019.

At Mango, only the month of January did not present a trend in rainfall data.

At Dapaong, 2 series out of 13 were found to be serially correlated at 95% (Figure 3). In fact, the month of February did not present any trend. Annually, the time series also did not display any trend.

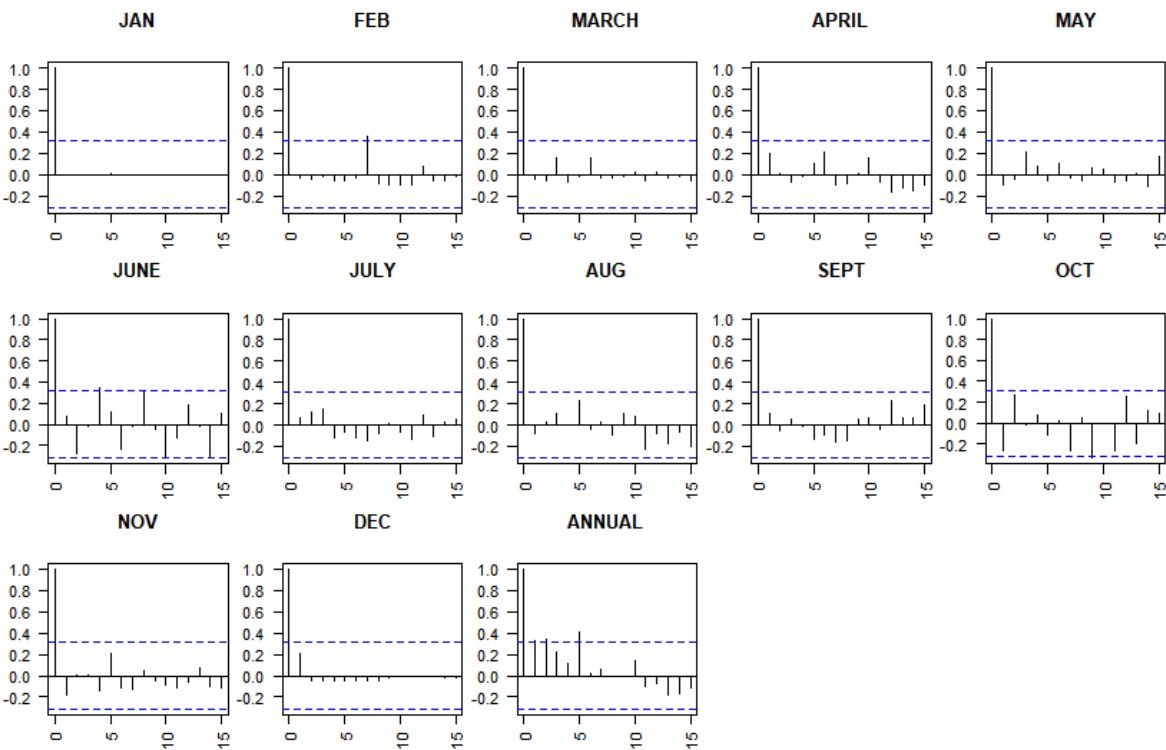

**Figure 3.** Autocorrelation test on monthly and annual rainfall (Dapaong station). Source: Dapaong meteorological service, 2019.

### 3.1.2. Annual Rainfall Trend Analysis

The result of total annual rainfall showed that 2001 and 1995 were the driest years in Mango and Dapaong, respectively, and 1999 and 2002 were the wettest years over the 1981–2019 period in the Mango and Dapaong stations, respectively in the Savannah region of Togo. The main rainy season contributed more than 50% to the annual rainfall. The majority of the yearly rainfall fell in May, June, July, August, September, and October (Tables 4 and 5). The months of November, December, and January were regarded as the driest months, contributing the least to the total yearly rainfall. This is a result of the Savannah region's primary rainy season, which lasts from May to October.

**Table 4.** Descriptive statistics of rainfall at Mango station.

| Period | Minimum (mm) | Maximum (mm) | Mean (mm) |
|---|---|---|---|
| January | 0 | 33.8 | 1.55 |
| February | 0 | 42.4 | 3.56 |
| March | 0 | 127.4 | 22.39 |
| April | 4.4 | 209.3 | 67.98 |
| May | 24.7 | 285.3 | 113.56 |
| June | 57.9 | 265.7 | 143.20 |
| July | 50.1 | 417.8 | 191.87 |
| August | 94.2 | 440 | 232.144 |

**Table 4.** *Cont.*

| Period | Minimum (mm) | Maximum (mm) | Mean (mm) |
|---|---|---|---|
| September | 4.8 | 342.9 | 188.54 |
| October | 0 | 197.5 | 72.19 |
| November | 0 | 42.2 | 3.26 |
| December | 0 | 47.1 | 1.21 |

Note: Source: Mango meteorological service, 2021.

**Table 5.** Descriptive statistics of rainfall at Dapaong station.

| Period | Minimum (mm) | Maximum (mm) | Mean (mm) |
|---|---|---|---|
| January | 0 | 98 | 2.54 |
| February | 0 | 70.1 | 3.93 |
| March | 0 | 220 | 20.53 |
| April | 0 | 196.8 | 63.35 |
| May | 22.5 | 306.9 | 104.8 |
| June | 68.1 | 247.1 | 147 |
| July | 54.7 | 984.4 | 211.3 |
| August | 4 | 784.7 | 272.2 |
| September | 3.5 | 348.8 | 189 |
| October | 0 | 182.8 | 60.3 |
| November | 0 | 51.1 | 4.44 |
| December | 0 | 29.8 | 0.95 |

Note: Source: Dapaong meteorological service, 2021.

The long-term annual mean total rainfalls for Mango and Dapaong were 1041.51 and 1080.35 mm, respectively. Most of total annual rainfall was less than the annual average, that is, 1041.51 at Mango station and 1080.35 at Dapaong station. At Mango, annual rainfall varied from 747.4 in 2001 to 1376.5 mm in 1998. The annual rainfall at Dapaong ranged from 765.3 in 1990 to 1756.6 in 2002. Over the 39 years considered in this study, 16 and 12 have their total rainfall higher than the average at Mango and Dapaong stations, respectively (Figure 4). This shows the lower rainfall experienced in the region.

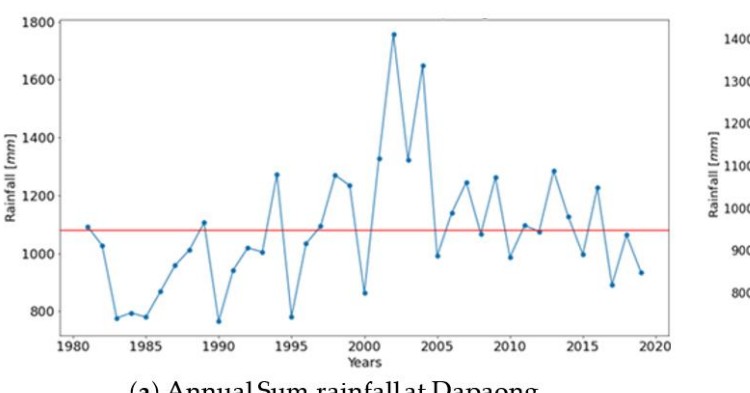 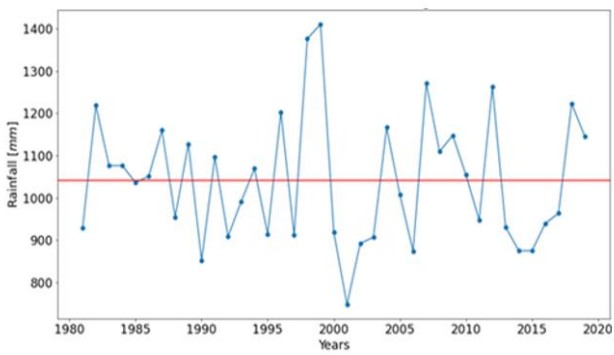

(**a**) Annual Sum rainfall at Dapaong  (**b**) Annual Sum rainfall at Mango

**Figure 4.** Annual rainfall at Mango and Dapaong. Source: Dapaong and Mango meteorological services, 2021.

In northern Togo there is a single rainy season in a year, which ranges from May to October (Walter, 1967 cited by Gadedjisso, 2021 [17]). The Figure 5 represent the graph of the twelve months average rainfall for the period 1981–2019. It shows one yearly peak in August for both Dapaong and Mango stations, which reveals the monomial pattern of rainfall in the study area.

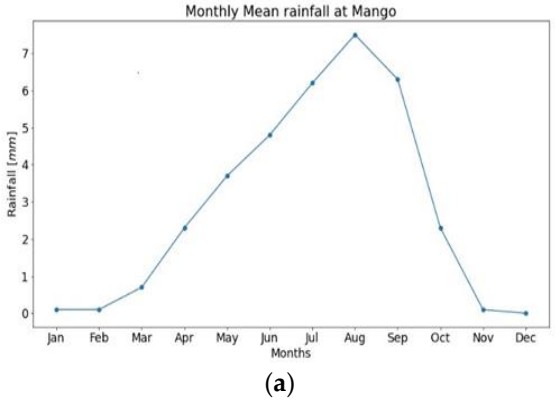

(**a**)

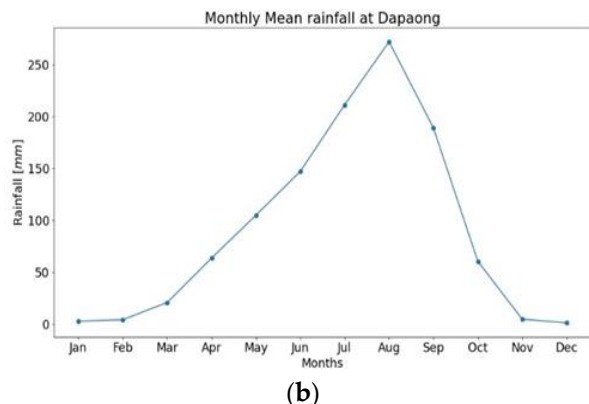

(**b**)

**Figure 5.** Monthly average rainfall at Mango (**a**) and Dapaong (**b**). Source: Dapaong and Mango meteorological services, 2021.

Mann–Kendall test (Table 6) results for monthly and yearly rainfall data had both positive and negative trends, with 95% confidence. Results indicated an increasing and significant trend in yearly rainfall at the Dapaong station, which is located in the northern part of the Savannah region ($p < 0.05$). Only the trend for May is statistically significant, though the months of April, May, June, August, September, and October showed an increasing tendency while July showed a decreasing trend. The monthly rainfall trends in April, May, July, August, September, and October at Mango station in the southern Savannah region were increasing, whereas the monthly rainfall trends in June were dropping. The months of May and September saw a major change in the pattern. Rainfall in Mango has declined ($-0.93$ mm/year), whereas it has increased (5.50 mm/year) in Dapaong.

**Table 6.** Man–Kendall test results of monthly and annual rainfall at Mango and Dapaong.

| Time Series | *p*-Value | | Sen's Slope | | MK | |
|---|---|---|---|---|---|---|
| | Dapaong | Mango | Dapaong | Mango | Dapaong | Mango |
| April | 0.122 * | 0.943 * | 0.526 | 0.040 | 0.174 | 0.009 |
| May | 0.022 * | 0.010 * | 1.092 | 1.513 | 0.258 | 0.289 |
| June | 0.371 * | 0.141 * | 0.726 | −0.830 | 0.101 | −0.166 |
| July | 0.561 * | 0.471 * | −0.700 | 0.816 | −0.066 | 0.082 |
| August | 0.699 * | 0.486 * | 0.406 | 0.7 | 0.045 | 0.080 |
| September | 0.095 * | 0.014 * | 1.733 | −2.487 | 0.188 | −0.275 |
| October | 0.432 * | 0.121 * | 0.540 | 0.173 | 0.089 | 0.173 |
| ANNUAL | 0.050 * | 0.701 * | 5.500 | −0.937 | 0.220 | −0.045 |

Notes: Source: Dapaong and Mango meteorological services, 2019. * Significant at 5%.

A modified Mann–Kendall test is used to look at trends when serial correlation has an impact on a series. Table 7 displays the findings of the modified Mann–Kendall test performed on the serially correlated series with a 95% confidence level.

**Table 7.** Modified Man–Kendall test results of monthly and annual rainfall at Mango and Dapaong.

| Time Series | Corrected Zc | | New *p*-Value | | Sen's Slope | | MMKY | |
|---|---|---|---|---|---|---|---|---|
| | **Dapaong** | **Mango** | **Dapaong** | **Mango** | **Dapaong** | **Mango** | **Dapaong** | **Mango** |
| April | 2.900 | 0.237 | 0.003 * | 0.812 * | 0.525 | 0.400 | 0.174 | 0.009 |
| May | 5.917 | 7.170 | 0.000 * | 0.000 * | 1.091 | 1.513 | 0.021 | 0.288 |
| June | 2.004 | −2.977 | 0.045 * | 0.002 * | 0.725 | −0.830 | 0.101 | −0.165 |
| July | −1.091 | 1.576 | 0.275 * | 0.114 * | −0.700 | 0.815 | −0.066 | 0.082 |
| August | 0.697 | 1.772 | 0.485 * | 0.076 * | 0.405 | 0.700 | 0.044 | 0.079 |
| September | 4.788 | −7.100 | 0.000 * | 0.000 * | 1.733 | −2.486 | 0.187 | −0.276 |
| October | 2.089 | 3.632 | 0.036 * | 0.000 * | 0.540 | 0.840 | 0.089 | 0.171 |
| ANNUAL | 2.551 | −1.236 | 0.010 * | 0.216 * | 5.500 | −0.937 | 0.219 | −0.039 |

Notes: Source: Dapaong and Mango meteorological services, 2019. * Significant at 5%.

The simple Mann–Kendall test and the modified Mann–Kendall test produced the same results in terms of trend in annual rainfall. This result corroborates the findings of Gadedjisso (2021) [17], who considered a time period from 1977 to 2012.

### 3.1.3. Annual Temperature Trend Analysis

The long term minimum average temperature were 22.2 and 22.9 in Mango and Dapaong, respectively, while the long term maximum average temperature were 35.5 and 33.7 in Mango and Dapaong, respectively. There was an increasing trend in the annual minimum temperature at Mango station and Dapaong station (Figure 6 and Table 8). A statistically significant trend was not found at Dapaong ($p > 0.05$), while at Mango ($p < 0.05$) a significant trend was found.

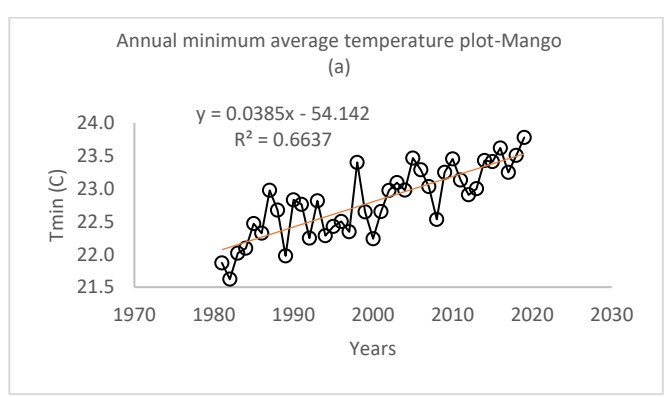
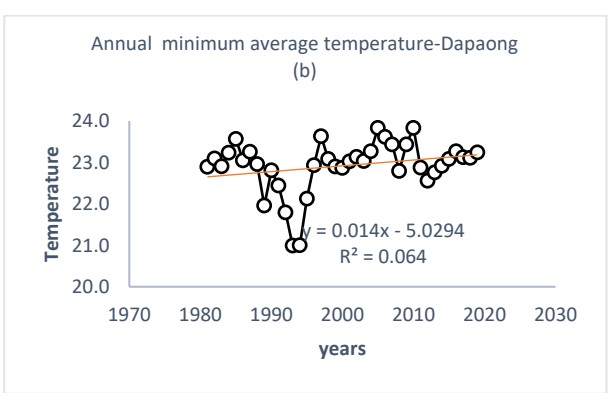

**Figure 6.** Annual minimum average temperature at Mango (**a**) and Dapaong (**b**). Source: Dapaong and Mango meteorological services, 2019.

**Table 8.** Mann–Kendall test results of annual and monthly minimum temperature—Dapaong.

| Time Series | *p*-Value (Two Tailed Test) | | Sen's Slope Estimate | | Man Kendall Statistic (S) | |
|---|---|---|---|---|---|---|
| | **Dapaong** | **Mango** | **Dapaong** | **Mango** | **Dapaong** | **Mango** |
| January | 0.133 * | 0.095 * | 0.029 | 0.032 | 0.169 | 0.188 |
| February | 0.240 * | 0.002 * | 0.027 | 0.053 | 0.132 | 0.340 |
| March | 0.008 * | 0.001 * | 0.029 | 0.058 | 0.299 | 0.417 |
| April | 0.529 * | 0.014 * | 0.006 | 0.030 | 0.071 | 0.275 |
| May | 0.536 * | 0.012 * | 0.004 | 0.033 | 0.070 | 0.280 |

**Table 8.** *Cont.*

| Time Series | *p*-Value (Two Tailed Test) | | Sen's Slope Estimate | | Man Kendall Statistic (S) | |
|---|---|---|---|---|---|---|
| | Dapaong | Mango | Dapaong | Mango | Dapaong | Mango |
| June | 0.203 * | 0.0001 * | 0.009 | 0.038 | 0.144 | 0.481 |
| July | 0.053 | 0.0001 * | 0.015 | 0.032 | 0.220 | 0.562 |
| August | 0.016 | 0.0001 * | 0.014 | 0.030 | 0.275 | 0.612 |
| September | 0.395 | 0.0001 * | 0.006 | 0.035 | 0.097 | 0.535 |
| October | 0.698 | 0.0001 * | 0.001 | 0.038 | 0.044 | 0.564 |
| November | 0.020 | 0.001 * | 0.026 | 0.056 | 0.263 | 0.357 |
| December | 0.255 | 0.040 * | 0.020 | 0.033 | 0.129 | 0.230 |
| ANNUAL | 0.143 | 0.0001 * | 0.008 | 0.040 | 0.164 | 0.625 |

Notes: Source: Dapaong and Mango meteorological services, 2021. * Significant at 5%.

Similarly, there was an increasing trend in the annual maximum temperature at Mango and Dapaong stations (Figure 7). The trend was statistically significant both at Mango ($p < 0.05$) and Dapaong stations. The increase in annual maximum temperature at Dapaong is contrary to the finding of Gadedjisso (2021) [17], who observed a decrease in annual maximum temperature at the same weather station over a period of 36 years (1977–2012). The minimum temperature increased by 0.04 and 0.008 °C per year in Mango and Dapaong, respectively. The maximum temperature increased by 0.03 and 0.02 °C per year in Mango and Dapaong, respectively. It is noticed from the results that the maximum temperature increased more rapidly than the minimum temperature at both weather stations; this may cause an extreme event, such as drought, (Table 9).

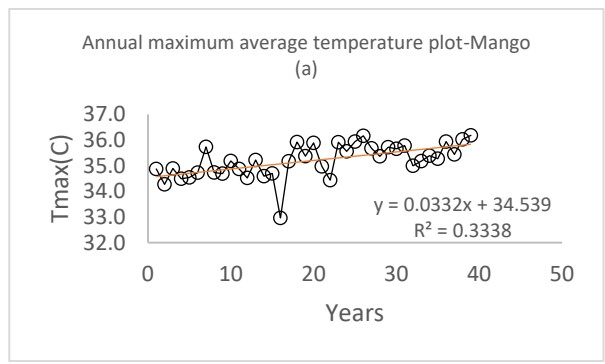 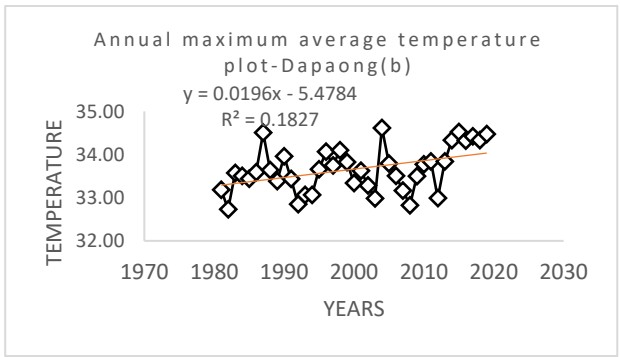

**Figure 7.** Annual maximum temperature at Mango (**a**) and Dapaong (**b**). Source: Dapaong and Mango meteorological services, 2019.

**Table 9.** Mann–Kendall test of annual and monthly maximum temperature—Mango.

| Time Series | *p*-Value (Two Tailed Test) | | Sen's Slope Estimate | | Man Kendall Statistic (S) | |
|---|---|---|---|---|---|---|
| | Dapaong | Mango | Dapaong | Mango | Dapaong | Mango |
| January | 0.270 * | 0.022 * | 0.017 | 0.040 | 0.125 | 0.255 |
| February | 0.164 * | 0.028 * | 0.023 | 0.040 | 0.157 | 0.246 |
| March | 0.023 * | 0.001 * | 0.030 | 0.048 | 0.260 | 0.389 |
| April | 0.309 * | 0.147 * | 0.014 | 0.027 | 0.114 | 0.162 |
| May | 0.467 * | 0.425 * | 0.012 | 0.014 | 0.082 | 0.089 |

**Table 9.** *Cont.*

| Time Series | *p*-Value (Two Tailed Test) | | Sen's Slope Estimate | | Man Kendall Statistic (S) | |
|---|---|---|---|---|---|---|
| | Dapaong | Mango | Dapaong | Mango | Dapaong | Mango |
| June | 0.255 * | 0.006 * | 0.040 | 0.034 | 0.129 | 0.304 |
| July | 0.167 * | 0.001 * | 0.015 | 0.040 | 0.156 | 0.397 |
| August | 0.244 * | 0.147 * | 0.010 | 0.014 | 0.133 | 0.162 |
| September | 0.594 * | 0.137 * | 0.005 | 0.010 | 0.060 | 0.166 |
| October | 0.771 * | 0.663 * | −0.005 | 0.004 | −0.033 | 0.049 |
| November | 0.011 * | 0.001 * | 0.040 | 0.039 | 0.286 | 0.378 |
| December | 0.018 * | 0.001 * | 0.043 | 0.056 | 0.267 | 0.386 |
| ANNUAL | 0.008 * | 0.0001 * | 0.022 | 0.032 | 0.295 | 0.455 |

Notes: Source: Dapaong and Mango meteorological services, 2019. * Significant at 5%.

3.1.4. Coefficient of Variation

Even when the means are drastically different from one another, the coefficient of variation, which measures the ratio of the standard deviation to the mean, is a helpful statistic for assessing how much variation there is between data sets.

The coefficient of variation (Table 10) in stations revealed that rainfall in the region has low inter-annual variability. The rainfall variability is CoV = 14% at Mango while it is CoV = 20% at Dapaong. This means that at Mango station the rainfall is less variable annually, and in Dapaong station the rainfall is moderately variable annually. The months of April, May, June, July, August, September, and October had a value of Coefficient of Variation higher than 0.20 (CoV > 20%); the rainfall is highly variable over these months at the two stations, while they correspond to the rainy season. This variability is one of the consequences of climate change, causing a serious challenge to farmers who are no longer able to monitor the growing cycle of their crops.

**Table 10.** Coefficient of Variation at the meteorological stations.

| Mango Station | | | Dapaong Station | | |
|---|---|---|---|---|---|
| Months | Stdev | CoV | Months | Stdev | CoV |
| Annual | 152.506 | 0.146 | Annual | 217.922 | 0.2017 |
| Jan | 6.798 | 4.375 | Jan | 15.688 | 6.161 |
| Feb | 9.441 | 2.647 | Feb | 12.829 | 3.262 |
| March | 29.342 | 1.310 | March | 39.254 | 1.912 |
| April | 48.14 | 0.708 | April | 39.074 | 0.61 |
| May | 50.845 | 0.448 | May | 50.047 | 0.478 |
| June | 54.783 | 0.383 | June | 54.313 | 0.372 |
| July | 75.284 | 0.392 | July | 143.15 | 0.677 |
| August | 72.261 | 0.311 | August | 126.07 | 0.463 |
| September | 73.853 | 0.392 | September | 73.224 | 0.388 |
| October | 51.601 | 0.715 | October | 41.836 | 0.694 |
| November | 7.884 | 2.417 | November | 10.454 | 2.353 |
| December | 7.541 | 6.205 | December | 4.8934 | 5.103 |

Note: Source: Dapaong and Mango meteorological stations, 2021.

### 3.2. Rainfall Anomaly Index

At both sites, the annual negative anomaly was greater than the annual positive anomaly. There was also considerable variability; a dry year was followed by two or three more dry years before being replaced by rainy years. Since 2000, there have been more negative anomalies in the Mango area, while in the Dapaong area, it was the positive anomalies that increased at the same time, Figure 8.

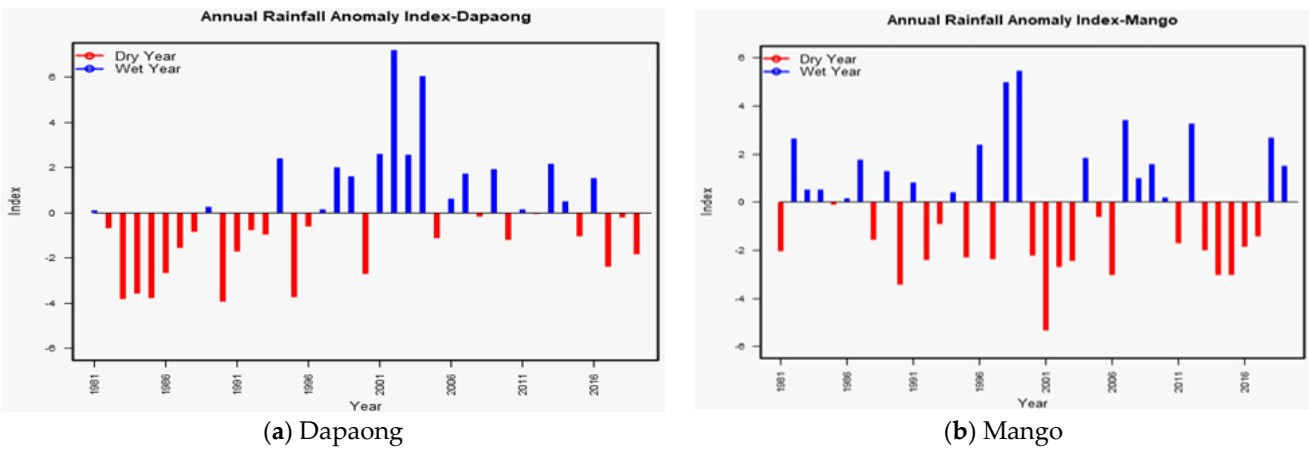

(**a**) Dapaong

(**b**) Mango

**Figure 8.** Annual rainfall anomaly index at Mango and Dapaong. Source: Dapaong and Mango meteorological services, 2019.

The result implies that due to more years of drought, crop production and food availability conditions of the region have been affected. Consistent with this result, Gadedjisso et al. (2021) [17] asserted that the variation in rainfall and temperature had a significant effect on the cereal crop yield in northern Togo.

The current analysis found that there has been variability in rainfall both within and between years. This variability may have an impact on the population in a different manner. Socioeconomically, this could alter their income since the main activity of the population is farming. Regarding the environment, this variability may exacerbate the land degradation process because, during the dry season, the soil is exposed to high temperatures, enabling the advancement of the desert. For a period of 39 years beginning in 1981, daily rainfall data for the districts of Mango and Dapaong was gathered and aggregated into monthly and yearly mean and totals. The drought and precipitation variability may increase the risk of food insecurity of poorer populations especially in rural settings. Therefore, it is clear that a more vulnerable environment will be favourable to migration phenomenon. This study took into account only two meteorological services for rainfall and temperature data collection. Further studies may consider all the meteorological stations in the Savannah region for such work.

In the Savannah region of Togo, the rate of deforestation is pronounced, leaving the soil bare and therefore increasing the albedo phenomenon. In fact, the soils in the Savannah region are dark. This characteristic of the soil enables an absorption of a large amount of rays which warms the earth and contributes to an increase in temperature. The increase in rainfall at Dapaong is in line with the findings of Gadedjisso (2021) [17], while the decrease in rainfall at Mango is contrary to his findings for the period from 1977 to 2012. The variability of the rainfall confirms the effects of climate change in the study area.

### 4. Conclusions

The trend in monthly and annual rainfall, monthly and annual minimum, and maximum temperature was investigated using the Mann–Kendall test and Sen's slope method. We used historical rainfall and temperature data from Mango and Dapaong meteorological stations over a period of 39 years starting from 1981. A modified version of the Mann–

Kendall test was performed on data that were serially correlated. The annual rainfall at Dapaong station exhibited an increasing trend, whereas the annual rainfall at Mango station showed a decreasing trend. There was an increasing trend in Tmin at both Mango and Dapaong. Similarly, there was an increasing trend in the Tmax at Mango and Dapaong stations. The increase in temperature is probably linked to the important phenomenon of albedo due to the characteristics of the soils. The Savannah region of Togo is getting hotter and the arid zone is spreading, leading to the desertification of the area. Migration phenomenon become crucial because of many factors including the high rainfall variability and the rain fed agricultural practices.

**Author Contributions:** Methodology, W.M.; Validation, E.K.; Visualization, W.L.F.; Writing—original draft, K.M.A.Y. All authors have read and agreed to the published version of the manuscript.

**Funding:** This study was funded by German's Ministry of Education under the German sponsored Programme—West African Science Centre for Climate Change and Adapted Land Use (WASCAL) rewrite this properly.

**Acknowledgments:** This paper has been funded by the International Climate Change Information and Research Programme (ICCIRP) and is part of the "100 Papers to Accelerate Climate Change Mitigation and Adaptation" initiative.

**Conflicts of Interest:** The authors have no relevant financial or non-financial interest to disclose. On behalf of all authors, the corresponding author states that there is no conflict of interest.

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
