# Peer review of "Time Series Analysis of Temperature and Rainfall in the Savannah Region in Togo, West Africa"

_water, doi:10.3390/w15091656_

Round 1

Reviewer 1 Report

The authors research the temporal of temperature and rainfall in Togo, which provide the reference for prediction of climate. However, there have some problems as follow.

1.In line 9, there have two abstract.

2.The English is a little poor. It is re-edited by native speaker. Some presentation is not clear enough. For instance, identify serially independent series in line 12, it is not clear enough. In line 32-33, exceeding global figures means exceeding global average temperature or average enhance temperature?In line 72-73, the regions's rainfall and temperature pattern are therefore impacted by this influence. Do the authors means rainfall and temperature pattern are impacted by poorest or climate variability?

3.In introduction, the authors illustrated much content of climate in Togo. However, it is lack of research background. For example, the influence of climate on population and urban heat.

4.In line 65-69, the authors illustrated this study targeted particularly the Savannah Region of the Togo. The exist research took the Northern part of the Togo as an example. What's the difference of temperature and rainfall between them . It should be introduced in discussion.

5.In line 120, Q1 is not explained in this research. What is the meaning of Q1 value?

6.In section 3.1.1 serial correlation analysis, the authors only illustrated the serially correlation simply. The authors should introduced the change of autocorrelation on rainfall and temperature in different months and years.

7.The symbol of * should be explained in table 6-9. For instance, * correlation is significant at the 0.05 level.

8.The authors used the historical data of Mango and Dapaong weather stations from 1981 to 2019 in line 10-11. While the data from figure 2 ,figure 2 and figure 6, table 4-10 are in 2021.

9.The innovation and outlook should be mentioned.

10.Some relevant references should be cited as follow.

Spatiotemporal relationship characteristic of climate comfort of urban human settlement environment and population density in China.Front.Ecol.Evol,2022,10:953725. doi:10.3389/fevo.2022.953725.

Relationships among local-scale urban morphology, urban ventilation, urban heat island and outdoor thermal comfort under sea breeze influence. Sustainable Cities and Society,2020,60:102289. doi:https://doi.org/10.1016/j.scs.2020.102289.

Contributions of sea-land breeze and local climate zones to daytime and nighttime heat island intensity. npj urban sustainability,2022,2:12. doi:https://doi.org/10.1038/s42949-022-00055-z.

Spatio-temporal evolution and factors of climate comfort for urban human settlements in the Guangdong-Hong Kong-Macau Greater Bay Area. Front.Environ.Sci,2022,10:1001064. doi:10.3389/fenvs.2022.1001064.

Reviewer 2 Report

Dear author/s

The manuscript is a pleasant reading. Although not a breakthrough research, it shows serious work developed. The methodology is well explained and the results are adequately analyzed. 

On page 2, there is a sentence that is a copy of a paper already published. This sentence must be deleted from the manuscript and rewritten appropriately. 

Please acknowledge my remarks throughout the text. 

I feel a lack of objectives for the study, it seems that you just wanted to assess if there have been changes in some climate variables. This is no novelty, although I understand the applicability (which needs to be explained). 

This leads to the conclusion that, because there is no defined objective, does not conclude. 

keep up the good work

Round 2

Reviewer 1 Report

The authors not solved my comments totally.

1.In introduction, the authors should introduce some existing research about temperature and rainfall, it called literature review. However, the authors only introduce the background and the characteristic of study area.

2.The authors only introduced the temperature and rainfall simply. In other words,the authors only illustrated the basic feature of temperatue and rainfall. The mechanism is not mentioned.

3.There are only have little content in page 13. Most of space is blank.

4.The conclusion is too short. The authors should concluded several points and explained each of them detailed.
